# EFFICIENT DYNAMIC STRUCTURED SPARSE TRAINING WITH LEARNED SHUFFLES

## ABSTRACT

Structured sparsity accelerates training and inference on modern GPUs, yet it still trails unstructured dynamic sparse training (DST) in accuracy. The shortfall stems from a loss of *expressivity*: whereas a dense layer can realise every possible mask obtained by choosing any $w$ active weights out of $n$, a fixed block, or $N{:}M$ layout explores only a subset of those possibilities. We propose to close this gap by learning, for each layer, a single permutation matrix jointly with the structured weight matrix. Applied to three canonical structures—block, $N{:}M$ and diagonals—we show that permutation-augmented DST (PA-DST) matches unstructured baselines (RigL, SET) at 90–95% sparsity on ImageNet-1K (ViT-B/16) and WikiText-103 (GPT-2), yet trains upto $1.21\times$ and infers up to $2.9\times$ faster. The results position *structure + learned permutation* as a sweet-spot between accuracy and efficiency.

## 1 INTRODUCTION

Over the years, deep neural networks (DNNs) have grown, and their performance on complex tasks has increased to and beyond human-level performance (Sparkes, 2023; Lykiardopoulou, 2023). However, the cost of training and inference for such large DNNs has increased drastically (Cottier, 2023). A principled way to curb this cost, while retaining dense level accuracy, is to exploit sparsity by removing unnecessary weights via pruning (Molchanov et al., 2016; Tanaka et al., 2020) or training directly in the sparse regime (Jaiswal et al., 2022; Zhang et al., 2023), often achieving similar algorithmic performance as dense models under comparable budgets (Frankle & Carbin, 2018; Blalock et al., 2020; Mostafa & Wang, 2019).

Weights are typically made sparse either *unstructurally*, by zeroing arbitrary individual parameters (Evci et al., 2020; Han et al., 2015), or *structurally*, by enforcing hardware-friendly patterns such as blocks or $N{:}M$ groups (Liu et al., 2019). Unstructured sparsity preserves accuracy at high sparsity levels but is hard to accelerate on modern accelerators due to irregular memory access and limited kernel support. Structured sparsity aligns well with vendor-optimized kernels, yet it commonly underperforms its unstructured counterpart—especially at extreme sparsity—because rigid patterns reduce the expressivity of the network. Moreover, many structured approaches still rely on dense backpropagation or lose structure under transposition, limiting practical training speedups (Lasby et al., 2023; Hubara et al., 2021).

**This work revisits structured sparsity from the angle of permutations.** Our key observation is that permutations help bridge the gap between the performance of structured and unstructured sparsity. We show the benefits of co-learning the *structure* and *permutations*. Concretely, for each sparsified layer $\ell$ with input $\mathbf{x}_\ell$ we learn a single permutation $\Pi_\ell$ and parameterize the layer as $y_\ell = S_\ell\big(\Pi_\ell\mathbf{x}_\ell\big)$, where $S_\ell$ obeys a fixed, accelerator-friendly structured sparsity pattern (e.g., block or $N{:}M$). While $S_\ell$ alone constrains the class of linear maps too severely, composing it with a learned permutation strictly enlarges the representable family (it recovers the classical structured model when $\Pi_\ell = I$ and extends it otherwise), thereby restoring much of the expressivity lost to structure. Importantly, the class is closed under transposition: $(S_\ell\Pi_\ell)^\top = \Pi_\ell^\top S_\ell^\top$, so the backward pass remains within the same permutation-structured family, enabling sparse-to-sparse acceleration in both forward and backward passes.

In addition to reintroducing expressivity, we *learn the sparsity itself during training*. Rather than fixing the position of non-zeros a priori, we dynamically adapt the structured mask of $S_\ell$ according to

Fig. 1: Overview of the PA-DST training approach. The training starts by initializing the target weight matrices with the desired sparse structure and a soft permutation matrix. As the training progresses, the soft permutation matrix starts moving towards a permutation matrix, and by the end of the training, we obtain the final sparse weight matrix and the permutation matrix. The permutation matrix is then used to reindex the activations of the layers so that an explicit matrix multiplication operation can be avoided during inference. More details can be found in Sec. 4.3.

task-driven criteria, coupling optimal non-zero positioning with the learned permutation $\Pi_\ell$. As we show in Sec. 6, this permutation-structured recipe delivers strong accuracy–speed trade-offs across architectures and sparsity regimes.

**This paper makes the following contributions:**

1. We present, **Permutation-Augmented DST** (PA-DST), a general layer formulation that combines any structured sparsity with a single learnt permutation matrix. We show that existing methods of learning permutations can be employed to transformer based models to improve their generalization performance.

2. We prove tight combinatorial bounds showing that the added permutation recovers the expressivity lost by structured sparsity as compared to dense and unstructured sparse models.

3. We demonstrate that with learnt permutations, structured sparse methods can achieve unstructured level accuracy across both vision (on ImageNet-1K) and language tasks (on WikiText-103), while achieving up to $1.21\times$ training and $2.9\times$ inference speedup at 90% sparsity for a ViT-B/16 network.

## 2 RELATED WORK

**Sparse training.** Classical pruning removes small-magnitude or low-salience weights from a pre-trained dense model and then fine-tunes the model. Despite strong compression, this approach still pays the dense pre-training cost (Molchanov et al., 2017; Cai et al., 2022; Lin et al., 2019; Lu et al., 2024). Lottery Ticket Hypothesis (LTH) (Frankle & Carbin, 2018) shows that dense nets contain sparse "winning tickets," but training sparse models from scratch can be more compute-intensive than dense training itself. To avoid dense pre-training, sparse-from-scratch methods fall into *static* and *dynamic* regimes. Static sparse training (SST) fixes a mask at initialization (e.g., Pixelated Butterfly (Dao et al., 2021)) but tends to underperform dynamic strategies at high sparsity. Dynamic sparse training (DST) updates connectivity during learning via prune-and-grow rules: SET prunes by magnitude and regrows randomly (Mocanu et al., 2018); MEST mixes magnitude and gradient signals (Yuan et al., 2021); RigL prunes by magnitude and regrows by gradient on missing connections (Evci et al., 2020); recent work explores topology-driven, gradient-free growth and N:M-style constraints for scalability (Zhang et al., 2024; 2023).

**Permutation learning and channel mixing.** Beyond fixed channel shuffles, *AutoShuffleNet (Lyu et al., 2020)* learns channel permutations *during training* by optimizing a Lipschitz-continuous penalty that drives a stochastic matrix toward a permutation. The *Kaleidoscope (Dao et al., 2020)* (K-matrix) framework provides a differentiable, expressive parameterization that includes permutations, and it has been used to *learn latent permutations* in permuted-image tasks within end-to-end training. Pool

et al. (Pool & Yu, 2021) use offline permutations to prune networks to a fixed N:M sparsity pattern. In contrast, we learn permutations jointly during training, and our method is pattern-agnostic, i.e., not tied to any specific structured sparsity, leading to more hardware-efficient sparse networks.

# 3 COMBINATORIAL EXPRESSIVITY VIA LINEAR REGIONS

The question we would like to answer is: **Why is there an accuracy gap between the generalization performance of a structured vs an unstructured sparse method?** Structured sparsity buys efficiency because accelerators can exploit regular patterns (blocks, $N{:}M$, diagonals). Yet those same patterns limit which directions each layer can "slice" the input space along. Unstructured sparsity, and of course dense networks, do not impose such directional restrictions and typically attain higher accuracy at the same sparsity. We make this precise by measuring expressivity via the *number of linear regions* (NLR) of ReLU networks and by asking: (i) how structure alone changes the classic depth-multiplicative region growth, and (ii) how a single learned permutation per layer restores it when depth is sufficient.

## 3.1 THEORETICAL SETUP & INTUITION

We consider a depth-$L$ feed-forward ReLU network with layers $\ell = 1, \ldots, L$:

$$z_\ell(x) = W_\ell\, a_{\ell-1}(x) + b_\ell, \qquad a_\ell(x) = \phi\big(z_\ell(x)\big), \qquad \phi(t) = \max\{t, 0\}, \qquad a_0(x) = x \in \mathbb{R}^{d_0}.$$

Here $x \in \mathbb{R}^{d_0}$ is the input; $a_{\ell-1}(x) \in \mathbb{R}^{d_{\ell-1}}$ the activation at layer $\ell-1$; $W_\ell \in \mathbb{R}^{d_\ell \times d_{\ell-1}}$ and $b_\ell \in \mathbb{R}^{d_\ell}$ the weight matrix and bias; $z_\ell(x) \in \mathbb{R}^{d_\ell}$ the pre-activation; and $a_\ell(x) \in \mathbb{R}^{d_\ell}$ the post-ReLU activation (with $\phi$ applied elementwise). We write $n_\ell := d_\ell$ for the width of layer $\ell$ and $(d_0, d_1, \ldots, d_L)$ for the layer dimensions. The activation pattern is constant on convex polyhedra; each maximal such set is a *linear region*.

**Hyperplane arrangements & expressivity via NLR.** Each neuron contributes an affine hyperplane that "slices" the affine subspace propagated to its layer. If, within that subspace, the hyperplanes are in *subspace-general position* (SGP)—i.e., their restricted normals are in general position—standard arrangement counts apply. We denote by $\mathrm{NLR}(f)$ the total *number of linear regions* of $f$. Classically, for dense ReLU networks with sufficiently wide layers, depth multiplies regions: each layer contributes a combinatorial factor that depends on the number of independent directions available, so $\mathrm{NLR}(f)$ grows multiplicatively with depth (Montúfar-style bounds (Montúfar et al., 2014)).

**A generic lower-bound template.** Let $k_\ell$ be the *effective dimension* at layer $\ell$ (the number of independent row-normal directions realizable inside the current region). Under SGP,

$$\mathrm{NLR}(f) \;\geq\; \prod_{\ell=1}^{L} \sum_{j=0}^{k_\ell} \binom{n_\ell}{j}. \tag{1}$$

All architectural effects reduce to identifying $k_\ell$. To reason uniformly across settings, we track an *accumulated span budget*

$$u_0 := 0, \qquad u_\ell := \min\{d_0,\, u_{\ell-1} + g_\ell\}, \tag{2}$$

where $g_\ell$ is the number of *fresh* (linearly independent) directions that layer $\ell$ can inject beyond those already spanned. The effective dimension obeys

$$k_\ell = \min\{\, n_\ell,\, h_\ell \,\}, \qquad \text{with } h_\ell \in \{u_{\ell-1},\, u_\ell\} \tag{3}$$

depending on whether the newly available directions are usable inside the current region. Subsequent subsections instantiate Eqn. 1–Eqn. 3 for dense matrices and for unstructured sparsity before turning to structured sparsity and permutations later.

## 3.2 DENSE MATRICES: RECOVERING CLASSICAL MULTIPLICATIVE GROWTH

Dense layers impose no directional restriction: any normal in the ambient subspace can be realized. Thus, there are no "fresh" directions to accumulate ($g_\ell = 0$) and the first layer already sees the full input subspace; we adopt the standard convention $u_0 = d_0$. From Eqn. 3,

$$k_\ell = \min\{n_\ell,\, u_{\ell-1}\} = \min\{n_\ell,\, d_0\} \quad \text{for all } \ell, \tag{4}$$

and plugging into Eqn. 1 yields

$$\mathrm{NLR}(f) \ \geq \ \prod_{\ell=1}^{L} \sum_{j=0}^{\min\{n_\ell, d_0\}} \binom{n_\ell}{j}. \tag{5}$$

If $n_\ell \geq d_0$ for all $\ell$, then $k_\ell = d_0$ layer-wise and each factor equals $\sum_{j=0}^{d_0} \binom{n_\ell}{j}$, giving the classical statement that *depth multiplies regions* (Montúfar et al., 2014).

### 3.3 Unstructured Sparsity

Unstructured sparsity does not impose intrinsic directional caps: with generic weights, the row normals of $W_\ell$ can span any $k \leq \min\{n_\ell, d_{\ell-1}\}$ directions inside the current subspace. Therefore, as in the dense case, we take $g_\ell = 0$ and $u_0 = d_0$, which by Eqn. 3 gives

$$k_\ell = \min\{n_\ell, \ u_{\ell-1}\} = \min\{n_\ell, \ d_0\}, \tag{6}$$

and the lower bound Eqn. 1 coincides with the dense bound:

$$\mathrm{NLR}(f) \ \geq \ \prod_{\ell=1}^{L} \sum_{j=0}^{\min\{n_\ell, d_0\}} \binom{n_\ell}{j}. \tag{7}$$

*Interpretation.* In the NLR lens, truly unstructured sparsity has the same depth-multiplicative expressivity as dense networks at a given width profile; any observed gap with structured sparsity arises from structural caps that reduce $k_\ell$ (analyzed in later sections) rather than from sparsity alone.

### 3.4 Structured Sparsity Without Mixing

We now turn to *structured* (axis-aligned) sparsity *without* any re-orientation across layers (e.g., no permutations/mixers). Let $\mathcal{A}_\ell \subset \mathbb{R}^{d_{\ell-1}}$ be the set of admissible row-normal directions at layer $\ell$ induced by the fixed pattern (diagonals, bands, blocks, or tied $N{:}M$ groups). Define the *directional rank cap* at layer $\ell$ by

$$r_\ell := \dim\big(\mathrm{span}(\mathcal{A}_\ell)\big) \ \leq \ d_{\ell-1}. \tag{8}$$

For the axis-aligned families we study, the orientation and admissible directions are the same at every depth, so $r_\ell = r_{\mathrm{struct}}$ is *constant* across layers. Set $s := \min\{d_0, r_{\mathrm{struct}}\}$. Because the admissible directions lie in the same $r_{\mathrm{struct}}$-dimensional coordinate subspace at every layer, the first layer can realize at most $s$ independent directions and no fresh directions can be injected later. Using the global template Eqn. 1, this yields

$$\mathrm{NLR}(f) \ \geq \ \prod_{\ell=1}^{L} \sum_{j=0}^{\min\{n_\ell, s\}} \binom{n_\ell}{j}. \tag{9}$$

*What this means.* Unlike dense or unstructured layers, $k_\ell$ is uniformly capped by $s$ independently of depth, so the per-layer factor in Eqn. 1 is bounded and depth-multiplicative growth stalls.

**Instantiations of $r_{\mathrm{struct}}$ for each sparsity structure:** For each sparsity structure, we instantiate $r_{\mathrm{struct}}$ as follows: for **Diagonal-**$K$, $r_{\mathrm{struct}} = K$, hence $s = \min\{d_0, K\}$; for **Block-**$B$, $r_{\mathrm{struct}} = B$, hence $s = \min\{d_0, B\}$; and for **N:M** with a tied group template, $r_{\mathrm{struct}} = \alpha d_0$ with $\alpha = N/M$, hence $s = \alpha d_0$. Substituting each resulting $s$ into Eqn. 9 gives the corresponding lower bound on $\mathrm{NLR}(f)$.

### 3.5 Structured Sparsity With Mixing

We now allow a *per-layer re-orientation*—a learned permutation or, more generally, any full-rank *mixer*—applied before the axis-aligned mask. Such mixing prevents successive layers from aligning to the same small coordinate subspace, so each structured layer can contribute up to $r_{\mathrm{struct}}$ *fresh* independent directions until the input dimension $d_0$ is saturated.

**Generic consequence of mixing.** Let the span budget evolve as $u_0 := 0$ and $u_\ell = \min\{d_0, u_{\ell-1} + r_{\mathrm{struct}}\}$, and use the effective-dimension cap $k_\ell = \min\{n_\ell, u_\ell\}$ in the master template Eqn. 1. This

yields the mixing-enabled lower bound

$$\text{NLR}(f) \geq \prod_{\ell=1}^{L} \sum_{j=0}^{\min\{n_\ell, u_\ell\}} \binom{n_\ell}{j}, \qquad u_\ell = \min\{d_0,\, u_{\ell-1} + r_{\text{struct}}\}. \tag{10}$$

*Meaning.* Each layer injects $r_{\text{struct}}$ new independent directions; the span budget grows additively and the dense per-layer factor is recovered after a short warm-up of

$$L_{\text{overhead}} = \left\lceil \frac{d_0}{r_{\text{struct}}} \right\rceil \text{ layers.} \tag{11}$$

The recipe above applies unchanged to each axis-structured family by substituting its $r_{\text{struct}}$: for **Diagonal-**$K$, $r_{\text{struct}} = K \Rightarrow u_\ell = \min\{d_0, u_{\ell-1} + K\}$ and $L_{\text{overhead}} = \lceil d_0/K \rceil$; for **Block-**$B$, $r_{\text{struct}} = B \Rightarrow u_\ell = \min\{d_0, u_{\ell-1} + B\}$ and $L_{\text{overhead}} = \lceil d_0/B \rceil$; and for **N:M**, $r_{\text{struct}} = \alpha d_0$ with $\alpha = N/M \Rightarrow u_\ell = \min\{d_0, u_{\ell-1} + \alpha d_0\}$ and $L_{\text{overhead}} = \lceil M/N \rceil$. With a single mixer per layer, axis-structured sparsity thus regains dense-like, depth-multiplicative expressivity at the same widths after an explicit, structure-dependent overhead.

**Why permutations (and what mixing suffices)?** Any *full-rank* per-layer mixer that varies across depth suffices for the theory above (e.g., permutations, co-prime stride shuffles, S-random interleavers, or very sparse full-rank transforms such as butterfly-style). We emphasize *permutations* because they are (i) parameter-free and invertible, (ii) cheap and friendly to accelerator memory layouts, and (iii) preserve axis-structure in $W_\ell$, allowing the same efficient kernels at inference. Empirically, learned permutations tend to outperform fixed random shuffles; the framework, however, only requires full rank and depth variation—not learnability per se. We discuss the practical consequences and predictions of our theory in detail in Apdx. C.

## 4 PERMUTATION–AUGMENTED DYNAMIC SPARSE TRAINING

This section presents **PA–DST**, our algorithm that learns (i) sparse weight values and positions (ii) one permutation matrix per layer, all under a fixed global sparsity budget. In Sec. 4.1, we describe how a linear layer formulation changes with the introduction of a permutation matrix. In Sec. 4.2 we elaborate on our approach to learn a permutation matrix in a differentiable manner. Lastly, in Sec. 4.3, we take the example of a ViT network to point out the changes when training with permutations.

### 4.1 LAYER FORMULATION

For a sparse weight matrix $W \in \mathbb{R}^{R \times C}$ we define permuted weight matrix $W' \in \mathbb{R}^{R \times C}$

$$W' = WP, \qquad P \in \mathcal{P}_d, \quad d = \min\{R, C\}, \tag{12}$$

where $P$ is a column permutation matrix. Here, $W$ is a sparse weight matrix where the positions of non-zeros are learnt during training, while adhering to a pre-defined structure. Row permutations can be applied to the weight matrices, but in our analysis, we don't find any significant difference in the algorithmic performance of the two permutations as shown in Sec. 6.4.

### 4.2 DIFFERENTIABLE PERMUTATION ESTIMATOR

Permutation matrices by nature are discrete, preventing direct gradient-based optimization methods. To ensure end-to-end differentiability, we require a method to learn permutation matrices through gradient descent. We adopt the permutation–learning formulation of AutoShuffleNet (Lyu et al., 2020) to enable end-to-end training. Rather than optimizing a discrete permutation matrix $P$, we learn a soft matrix $M \in \mathbb{R}^{N \times N}$ constrained to the Birkhoff polytope (doubly–stochastic matrices), and decode a hard permutation at evaluation time. Following Lyu et al. (2020), we couple our task loss with an exact, Lipschitz-continuous $\ell_1 - \ell_2$ row/column penalty that drives a doubly–stochastic $M$ toward a true permutation:

$$\mathcal{L} = \mathcal{L}_{\text{task}} + \lambda P(M) \quad \text{subject to} \quad M \geq 0,\ M\mathbf{1} = \mathbf{1},\ M^\top \mathbf{1} = \mathbf{1}, \tag{13}$$

$$P(M) = \sum_{i=1}^{N} \Big( \|M_{i:}\|_1 - \|M_{i:}\|_2 \Big) + \sum_{j=1}^{N} \Big( \|M_{:j}\|_1 - \|M_{:j}\|_2 \Big). \tag{14}$$

As shown by Lyu et al. (2020), for matrices satisfying the doubly–stochastic constraints, $P(M) = 0$ if and only if $M$ is a permutation. In our setting, this penalty provided the most stable training among soft-permutation alternatives.

### 4.3 Permutation Matrices during training and inference

For this work, we are interested in looking at transformer-based neural networks. So, for a general transformer block, such as in a ViT-B/16, we sparsify the output projection linear layers of the multi-headed (MHA) attention block and the fully connected layers in the feedforward (MLP) blocks. This is in line with previous work in the literature (Lasby et al., 2023; Tyagi et al., 2025). We describe below in detail how the abovementioned layers are transformed.

**MHA output projection.** Let $H \in \mathbb{R}^d$ be the concatenated head output and $W_O \in \mathcal{S} \subset \mathbb{R}^{d \times d}$ the structured–sparse output projection. We learn a column permutation $P_O$ and the **forward pass** can be written as:

$$o = W_O P_O H = W_O a, \qquad a := P_O H. \tag{15}$$

During **inference**, we know the learned permutation next to the attention output linear: $W_O$ is the *output projection* (from $d$ to $d$), and $P_O$ permutes the concatenated head output $H \in \mathbb{R}^d$. A direct implementation would compute $o = W_O (P_O H)$. Instead of multiplying by $P_O$, we precompute an index map $\ell_O : [d] \to [d]$ such that $(P_O H)_i = H_{\ell_O(i)}$, and we re-index during head concatenation (i.e., we write the concatenated buffer in permuted order, or equivalently read it with permuted indices), then run the usual sparse GEMM

$$o_k = \sum_{i=1}^{d} W_O[k, i] \, H_{\ell_O(i)}. \tag{16}$$

This keeps $Q, K, V$ and the softmax unchanged, adds no extra matmul or copy, and is cheaper than a permutation multiply.

**FFN / MLP (both linears sparsified; permutations absorbed at inference).** Let $W_\uparrow \in \mathcal{S} \subset \mathbb{R}^{d_{\mathrm{ff}} \times d}$ denote the *expansion* (up-projection) and $W_\downarrow \in \mathcal{S} \subset \mathbb{R}^{d \times d_{\mathrm{ff}}}$ the *contraction* (down-projection) matrix in the FFN. We learn one column permutation for each sparsified linear: $P_\uparrow \in \{0, 1\}^{d \times d}$, and $P_\downarrow \in \{0, 1\}^{d_{\mathrm{ff}} \times d_{\mathrm{ff}}}$ as a column permutation of $W_\downarrow$. With $\sigma(\cdot)$ being the activation and $\mathrm{LN}(\cdot)$ being the layer norm, the forward pass is

$$y = W_\downarrow P_\downarrow \, \sigma\big(W_\uparrow P_\uparrow \mathrm{LN}(x)\big) . \tag{17}$$

Now, at **inference**, multiplying by permutation matrices involves extra computation. Instead of performing extra multiplications, we remove them by *re-indexing*. We precompute index maps $\ell_\uparrow : [d] \to [d]$ and $\ell_\downarrow : [d_{\mathrm{ff}}] \to [d_{\mathrm{ff}}]$ so that $(P_\uparrow x)_j = x_{\ell_\uparrow(j)}$ and $(P_\downarrow h)_i = h_{\ell_\downarrow(i)}$. Forward pass is:

$$u_i = \sum_{j=1}^{d} W_\uparrow[i, j] \, x_{\ell_\uparrow(j)}, \qquad h = \sigma(u), \qquad y_k = \sum_{i=1}^{d_{\mathrm{ff}}} W_\downarrow[k, i] \, h_{\ell_\downarrow(i)}. \tag{18}$$

No explicit $P_\uparrow$ or $P_\downarrow$ is formed at runtime; we only change which indices we read or write inside the existing kernels. This is computationally cheaper than multiplying by permutation matrices because a permutation multiply performs a shuffle, which would add an extra pass over memory and an extra kernel launch, whereas re-indexing costs only a small lookup and a few integer address operations per element, with no extra matrix multiplies, and unchanged structured-sparse GEMM multiplication.

## 5 Experimental Setup

The aim of our experiments is to show the benefits of learning permutations during dynamic sparse training of neural networks. We aim to show the benefits of different modalities (vision and language), model types (MLP and attention), and sparsity regimes.

**Evaluation.** We assess the generalization ability of all structured sparse methods with and without permutations using Top-1 accuracy for vision tasks and perplexity (PPL) for language tasks. Additionally, we measure the training and inference times for each method at varying sparsity levels through end-to-end execution, and assess the overheads associated with permutations-based training.

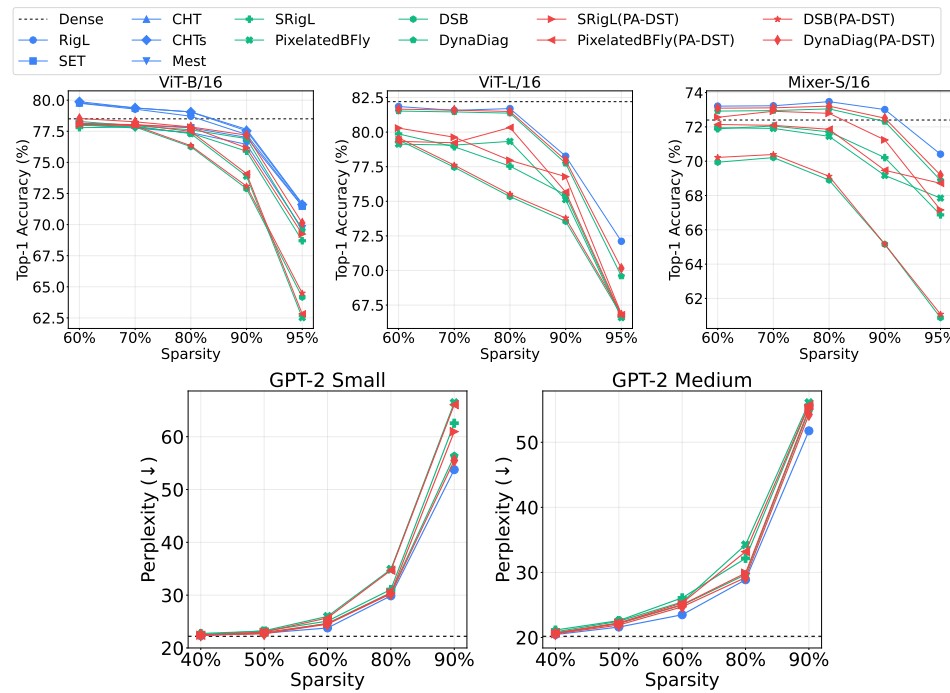

Fig. 2: Accuracy vs sparsity of networks listed in Sec. 5. The plots in blue, green, and red correspond to unstructured, structured, and structured + permutations-based (PA-DST) methods, respectively. We see that a permutation helps improve the generalization performance (red plots are above the green plots). Moreover, permutations help bridge the gap between unstructured and structured sparse methods (structures + permutations, in red plot, reach close to the unstructured methods, in blue).

**Baselines.** We compare the performance of the structured DST with & without permutation against the following approaches: A **Dense** network is trained with 0% sparsity in the weight matrices. For **Unstructured DST**, we use methods from the literature such as RigL (Evci et al., 2020), Mest (Yuan et al., 2021), SET (Mocanu et al., 2018), CHT (Zhang et al., 2024), and CHTs (Zhang et al., 2025) to produce unstructured sparse neural networks. For **Structured DST**, we use SRigL (Lasby et al., 2023), DSB (Jiang et al., 2022), and DynaDiag (Diag Sparsity) (Tyagi et al., 2025) to train networks with structured sparsity via DST, and we use PixelatedBFly (Dao et al., 2021) to compare against a structured **Static Sparse Training (SST)** method. Finally, for **Structured Sparse Training + Permutations**, we apply either random or learned permutations to the above structured DST methods to observe the impact on the network's generalization performance. In the random case, a permutation is generated before the start of training and applied to the network.

# 6 RESULTS

We present our main results, comparing the performance of various structured DST methods with and without permutations for vision (Sec. 6.1) and language (Sec. 6.1.1) tasks. We then compare the runtime of all sparse methods (training and inference time on GPUs) in Sec. 6.2 to understand the overheads associated with permutation-based learning. We wrap up this section by looking at the learned permutations in more detail (Sec. 6.3) and showing the results of ablation studies in Sec. 6.4.

## 6.1 ACCURACY VERSUS SPARSITY: VISION EXPERIMENTS

**Setup.** We evaluate the two architectures for vision tasks on ImageNet-1K (Deng et al., 2009) dataset. We train two variants of Vision Transformers (ViTs) (Dosovitskiy, 2020), ViT-B/16 and ViT-L, to study scalability aspects of sparsity. We also train Mixer-S/16 variant of MLP-Mixers (Tolstikhin et al., 2021) for our studies. We test all the sparse training methods (with and without permutations)

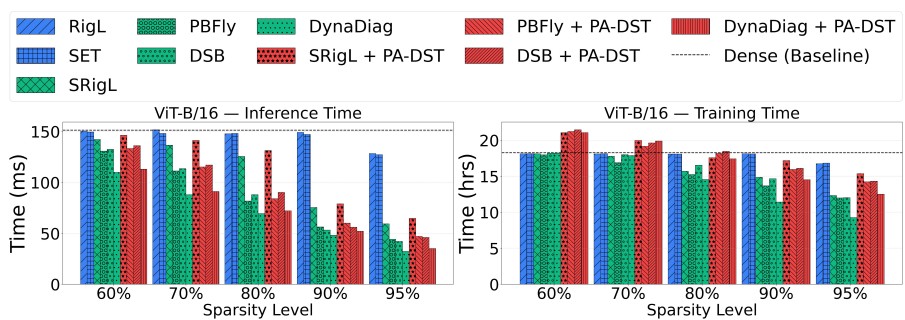

Fig. 3: Impact of permutations on the training and inference time for a ViT-B/16 model. We observe that there is a 3.16% - 8.69% overhead related to permutations during inference for the obtained accuracy gains. Even for training, learning permutations increases the overall training time, but at higher sparsity levels, we still obtain speedup compared to a dense model for all structured sparsities. Training time for GPT models can be found in Tbl. 5.

at 60%, 70%, 80%, 90%, and 95% sparsity in the targeted layers in the networks. Details of which layers are sparsified can be found in Apdx. C.5.

**Results:** Fig. 2a) - c) shows a summary of all our results, comparing the performance of the three networks at varying levels of sparsities, reporting the average of three runs. The results show that learnt permutations improve the generalization performance of the structured sparse networks, helping bridge the gap with unstructured sparsity. For instance: On ViT-L/16, at 70% sparsity, we obtain RigL level accuracies with DynaDiag + PA-DST. We see an improvement in accuracy across all structured sparsity patterns with an increase of 1.22% for SRigL at 90% sparsity (ViT-L). The raw values, along with comparisons with random permutations, can be found in Tbl. 11.

### 6.1.1 PERPLEXITY VERSUS SPARSITY: LANGUAGE EXPERIMENTS

**Setup:** For language tasks, we evaluate on the WikiText-103 dataset using two variants (Small and Medium) of GPT-2 Radford et al. (2019) at varying sparsities $s \in \{40\%, 50\%, 60\%, 80\%, 90\%\}$. We sparsify both the attention and the MLP layers of the model.

**Results.** We train GPT2-Small and GPT2-Medium Radford et al. (2019) from scratch on WikiText-103 dataset and show the resulting performance in Fig. 2d) and e). We report the average PPL obtained from *two* runs. We see that structured sparse models benefit from learning permutations, achieving similar PPL to that of unstructured sparse networks. Notably, PA-DST-based methods demonstrate increasingly significant improvements over no-permutation-based baselines. For example, at 80% sparsity on GPT2-Medium, SRigL + PA-DST achieves a perplexity of 2.27 points lower than that just SRigL while remaining within 0.98 points of the performance ceiling established by RigL. The raw values, along with comparisons with random permutations, can be found in Tbl. 12.

### 6.2 TRAINING AND INFERENCE COST

Learning permutations means extra computation, and hence it is essential to understand the overheads associated with the same. We measure the inference and training wall clock times of various structured sparse training approaches with and without permutations.

**Setup.** We use the best available libraries to accelerate the execution of structured sparsities mentioned in Sec. 5, and we use cuSparse (Naumov et al., 2010) to execute all the unstructured sparsities. We estimate the execution times for SRigl (Lasby et al., 2023) as per their methodology and are unable to run end-to-end training and inference due to lack of support in PyTorch. Whereas, for **DSB and PixelatedBFly**, we use the Triton based library package from PixelatedBFly (Dao et al., 2021) to accelerate inference for both and training for PixelatedBFly (DSB lacks support to integrate the kernels with the training pipeline). And lastly, we use the provided CUDA kernels implemented to accelerate **DynaDiag's** execution on the GPUs.

**Results.** In Fig. 3, we compare the inference and training times of models listed in Sec. 6.1 and Sec. 6.1.1. We see that using the approach mentioned in Sec. 4.3, we can re-index the output of a layer instead of explicitly applying a permutation matrix, leading to an overhead of just up to **8.69%** as compared to without permutation baselines at all sparsity levels. Even with the overhead, we can achieve an inference speedup as high as 2.9× with DynaDiag at 90% sparsity. Whereas, during training, we see an increase in training time for all structured sparse methods due to extra computation required for learning the permutations. We elaborate in Apdx. C.2 overheads & our approach to early stopping the permutation learning by tracking the loss and heuristically determining a stopping point. However, we can see that at higher sparsities, even with permutations, it is possible to get better training time than dense and unstructured sparse models. For example, at 95% sparsity, the training times for SRigL, PBFly, DSB, and DynaDiag are 1.09×, 1.18×, 1.13×, and 1.23× that of RigL, respectively. We can also see that DynaDiag is the fastest to train the sparse models, and this can be attributed to the transposable nature of the sparse pattern (Tyagi et al., 2025), which helps accelerate the backward pass.

### 6.3 Learned Permutations

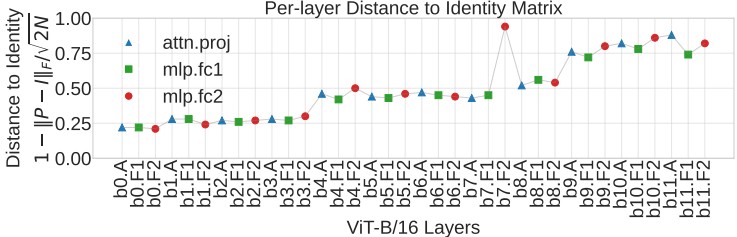

Fig. 4: Distance of permutations to an identity matrix (no permutation) for a ViT-B/16 at 90% sparsity trained with DynaDiag. We see that with depth, the learned permutations move closer to an identity matrix, signalling less shuffling with depth. A: Attention block; F: Fully Connected; b:blocks

To understand what the learned permutations look like, we take the example of a ViT-B/16 trained with DynaDiag at 90% sparsity and plot the normalized distance between the learned permutations and an identity matrix. The identity is the natural no-permutation baseline, so measuring proximity to it directly measures how much reordering a layer learns. We use a width–invariant, normalized Frobenius metric for a permutation matrix $P \in \mathbb{R}^{N \times N}, \delta(P) = 1 - \frac{\|P-I\|_F}{\sqrt{2N}} \in [0, 1]$. The result is shown in Fig. 4. We observe two primary trends: 1) across depths, early blocks display the lowest distance to the identity matrix, conveying that they learn stronger reindexing, and later layers, permutation matrices are closer to an identity matrix, meaning weaker shuffles. 2) attn projections tend to remain more permuted than MLP layers, which aligns more with an identity matrix with depth.

### 6.4 Ablation study

**Row vs Col Permutations.** Training structured sparse networks with col or row permutations yield networks with similar generalization performance. Tbl. 10 shows the results of training a structured sparse network col vs row permutation matrix for a ViT-B/16 network on ImageNet-1K. This means that instead of formulating a linear layer as $y = WPx$, we train instead with $y = PWx$, while keeping all other hyperparameters identical. We observe that there is **no significant difference** between their generalization performance.

## 7 Conclusion

We theoretically show that *learning one permutation matrix per layer* restores the expressivity lost in structured sparsity while preserving accelerator–friendly layouts. We back these claims with experimental results showing that training structured sparsity with the resulting PA-DST algorithm attains unstructured-level accuracy while still realizing speed-ups during training and inference.

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

# Appendix

## A  MAPPING DENSITY TO PATTERN PARAMETERS

Given a target per-layer density $\delta \in (0, 1]$ and input size $n_{\text{in}}$, we choose the smallest feasible integers so that per-row nnz $\approx \delta\, n_{\text{in}}$:

$$K = B = \text{round}(\delta\, n_{\text{in}}), \qquad 2b+1 = \text{nearest odd to } \delta\, n_{\text{in}}, \qquad \alpha = \frac{N}{M} = \delta \ \text{ (tied } N{:}M).$$

For non-cyclic bands/diagonals, use wrap-around (or adjust a few edge rows by $\pm 1$ nnz) so the total nnz matches the target. In our ViT-L/16 surrogate at $\delta = 0.05$: $n_{\text{in}}{=}1024 \Rightarrow K{=}B{=}51$, $2b{+}1{=}51$; $n_{\text{in}}{=}4096 \Rightarrow K'{=}B'{=}205$, $2b'{+}1{=}205$; tied $N{:}M$ uses $\alpha = 0.05$ throughout.

## B  WORKED-EXAMPLE CALCULATIONS (DETAILS)

Using the master bound equation 1 and span update $u_\ell = \min\{d_0, u_{\ell-1} + r_{\text{struct}}(n_{\text{in},\ell})\}$ with $d_0 = 1024$, the alternating widths yield per-block gain

$$r_{\text{pair}} = r_{\text{struct}}(1024) + \min\{r_{\text{struct}}(4096),\, d_0\} = 51 + 205 = 256.$$

Thus $u_{2t} = \min\{1024, 256\,t\}$ and dense-like factors are guaranteed once $u_{2t} = 1024$, i.e., $t = \lceil 1024/256 \rceil = 4$ blocks ($\approx 8$ layers). Without mixing, $u_\ell \equiv 51$ and the per-layer factor remains strictly below dense for all depth.

## C  PRACTICAL CONSEQUENCES & PREDICTIONS OF OUR THEORY

We now analyze the impact of our theory via a concise worked example where *all* structured families operate at (approximately) the same sparsity level. To stay within our ReLU/MLP framework, we use an MLP surrogate whose layer sizes are motivated by ViT-L/16, which has 24 encoder blocks (307M params). Each block contains a two-layer FFN with $d_{\text{model}}{=}1024$ and $d_{\text{ff}}{=}4096$. We analyze the MLP that stacks these FFN layers in order,

$$\underbrace{1024 \to 4096 \to 1024}_{\text{block 1 FFN}} \to \underbrace{1024 \to 4096 \to 1024}_{\text{block 2 FFN}} \to \cdots \text{ (24 blocks)},$$

yielding a depth-$L{=}48$ ReLU-MLP used solely to set widths for the master bound equation 1. At **95% sparsity** (density $\delta{=}0.05$), the structural caps (Sec. 3.4) are

$$r_{\text{struct}}(1024) = 51, \qquad r_{\text{struct}}(4096) = 205,$$

for Diagonal-$K$, Banded-$b$ (with $2b{+}1$ odd), Block-$B$, and tied $N{:}M$ with $\alpha = N/M = 0.05$ (mapping in App. A). *Consequences.* Without mixing, all axis-structured families stall at $k_\ell \leq 51$ (no multiplicative growth). With one mixer per layer (permutations suffice), the span budget grows additively; because widths alternate $1024 \leftrightarrow 4096$, each block contributes $51{+}205 = 256$ fresh directions toward $d_0{=}1024$, so dense-like per-layer factors are guaranteed after

$$\lceil 1024/256 \rceil = 4 \text{ blocks } (\approx 8 \text{ layers}).$$

In this setting, the structured families share the *same* catch-up point (4 blocks) when mixing is used.

### C.1  COMBINATORIAL EXPRESSIVITY EXAMPLE

Take $d_0 = 4$, equal widths $n_\ell = 8$, and $L = 3$ layers.

1. **Dense / Unstructured (RigL/SET-like)**: $k_\ell = \min\{n_\ell, d_0\} = 4$ at every layer. Per-layer factor $\sum_{j=0}^{4} \binom{8}{j} = 1 + 8 + 28 + 56 + 70 = 163$. Hence $NLR \geq 163^3$.

2. **Block-$B$ without permutations**, $B = 2$: $k_\ell = \min\{n_\ell, B\} = 2$ at every layer (no new directions with depth). Per-layer factor $1 + 8 + 28 = 37$. Hence $NLR \geq 37^3$.

3. **Block-$B$ with a learned permutation each layer**, $B = 2$ ($r_s = 2$): $u_0 = 0 \to u_1 = 2$ and $u_2 = 4$; thereafter $u_\ell = 4$. Per-layer factors: layer 1 gives 37, layers 2 and 3 give 163 each. Thus $NLR \geq 37 \cdot 163 \cdot 163$. After a one-layer overhead, the per-layer factor *matches dense*.

This concretely illustrates the phenomenon: structure alone stalls multiplicative growth; adding permutations restores it after a short, explicit warm-up in depth.

Table 1: **Lower bounds summary** (instantiate equation 1). Here $d_0$ is the input dimension, $n_\ell$ the width, and $r_s \in \{K, 2b+1, B\}$ the per-layer structural cap for diagonal/banded/block. For tied $N{:}M$, $\alpha = N/M$.

| Setting | Effective $k_\ell$ | Span recursion $u_\ell$ | Depth overhead |
|---|---|---|---|
| Dense | $\min\{n_\ell, d_0\}$ | $u_\ell = d_0$ | 0 |
| Unstructured DST (free masks) | $\min\{n_\ell, d_0\}$ | $u_\ell = d_0$ | 0 |
| $N{:}M$ (free supports) | $\min\{n_\ell, d_0\}$ | $u_\ell = d_0$ | 0 |
| $N{:}M$ (tied template) | $\min\{n_\ell, \alpha u_{\ell-1}\}$ | $u_\ell = u_{\ell-1}$ | – (stalls) |
| Diagonal-$K$ (no perm) | $\min\{n_\ell, K\}$ | $u_\ell = \min\{d_0, K\}$ | – (stalls) |
| Banded-$b$ (no perm) | $\min\{n_\ell, 2b+1\}$ | $u_\ell = \min\{d_0, 2b+1\}$ | – (stalls) |
| Block-$B$ (no perm) | $\min\{n_\ell, B\}$ | $u_\ell = \min\{d_0, B\}$ | – (stalls) |
| Diagonal-$K$ + permutation | $\min\{n_\ell, u_\ell\}$ | $u_\ell = \min\{d_0, u_{\ell-1} + K\}$ | $\lceil d_0/K \rceil$ |
| Banded-$b$ + permutation | $\min\{n_\ell, u_\ell\}$ | $u_\ell = \min\{d_0, u_{\ell-1} + 2b+1\}$ | $\lceil d_0/(2b+1) \rceil$ |
| Block-$B$ + permutation | $\min\{n_\ell, u_\ell\}$ | $u_\ell = \min\{d_0, u_{\ell-1} + B\}$ | $\lceil d_0/B \rceil$ |

## C.2 PERMUTATION LEARNING SCHEDULE & OVERHEAD

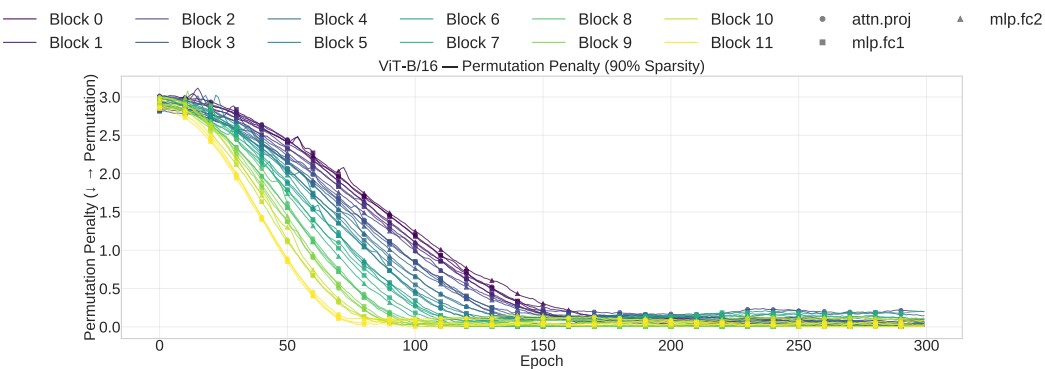

Fig. 5: Tracking the loss of permutation matrices based on Eqn. 14. We plot the loss for every 10 training epochs.

### C.2.1 PERMUTATION LEARNING SCHEDULE

While training, we can either learn the permutations till the end of the training or find a way to stop early without losing out on optimal permutations. From our experiments, we find that it is useful to monitor the loss of the soft-permutation matrix. For our training, we set a threshold loss value $\delta$ at which point we go from a soft to a hard permutation matrix for that particular layer. This means that instead of carrying out a multiplication operation for that layer, we can instead do re-indexing, which is a much cheaper operation.

We show in Fig. 5 the loss associated with each permutation matrix when training a ViT-B/16 network at 90% sparsity with DynaDiag. We can see that the loss decreases drastically and saturates after the knee in the plot. Moreover, we see a clear trend that earlier permutation matrices take longer to reach this knee point. For the given experiment, we set the $\delta = 0.22$ and we show in Fig. 6 when each layer reaches that threshold, and we stop training the permutation matrix corresponding to that layer.

### C.2.2 OVERHEADS AND ADDITIONAL RESULTS

The tables quantify the training overhead associated with various permutation methods. Specifically, Tbl. 2 and Tbl. 3 detail the memory overhead in gigabytes and as a percentage for the GPT-2 Small model, using Diagonal and N:M sparsity, respectively. Tbl. 4 presents a similar memory overhead analysis for the ViT-B/16 model, but at higher 90% and 95% sparsity levels. Finally, Tbl. 5 expands this analysis for the GPT-2 Medium model, showing the overhead for both training time (in hours) and memory. Across all tables, we compare permutation strategies such as AutoShufflePerm and

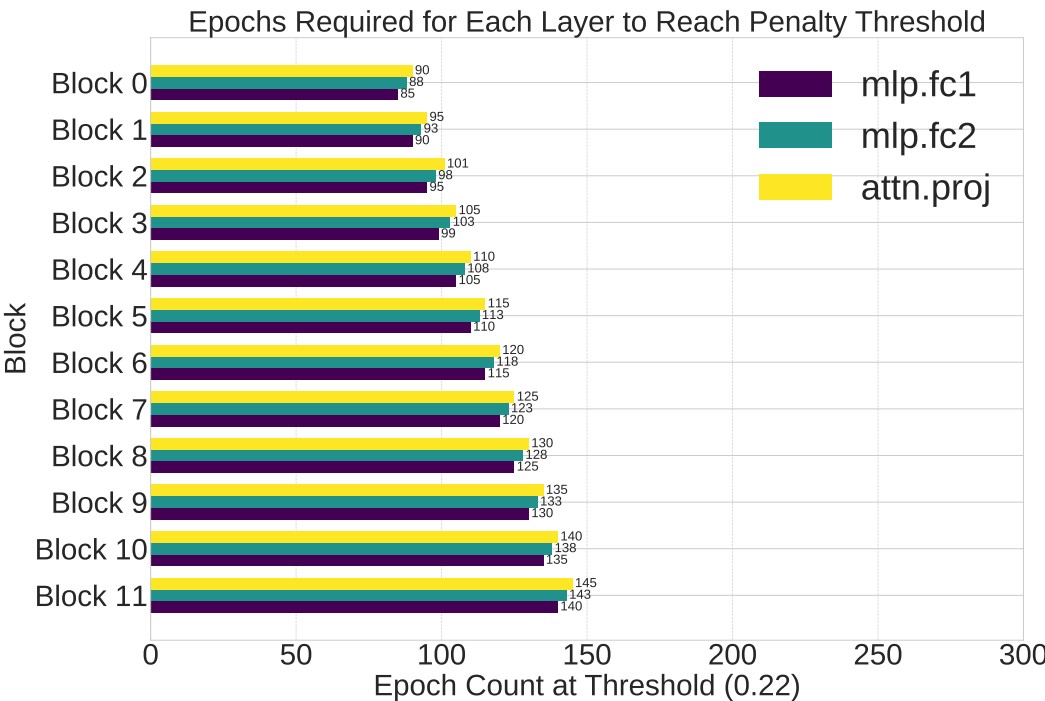

Fig. 6: We show the epoch count of each layer in a ViT-B/16 network when the corresponding permutation loss hits the threshold. We can see that the cutoff epoch varies drastically acorss the network which we use to our advantage to reduce the training time.

the KaleidoscopePerm against their corresponding non-permuted baselines to clearly illustrate the computational trade-offs.

Table 2: Memory overhead of permutation methods for GPT-2 Small with Diagonal sparsity on WikiText-103. The overhead percentage is calculated relative to the DynaDiag method.

| Model | Method | Memory (GB) @ 60% | % Overhead | Memory (GB) @ 80% | % Overhead |
|---|---|---|---|---|---|
| | Unstructured | 133.64 | - | 132.12 | - |
| | DynaDiag | 137.32 | - (Baseline) | 132.82 | - (Baseline) |
| GPT-2 Small (DynaDiag) | + FixedRandPerm | 139.21 | **+1.38%** | 134.09 | **+0.96%** |
| | + PA-DST | 156.17 | **+13.73%** | 145.41 | **+9.48%** |
| | + KaleidoscopePerm | 141.27 | **+2.88%** | 137.43 | **+3.47%** |

Table 3: Memory overhead of permutation methods for GPT-2 Small with SRigL on WikiText-103. The overhead percentage is calculated relative to the SRigL method.

| Model | Method | Memory (GB) @ 60% | % Overhead | Memory (GB) @ 80% | % Overhead |
|---|---|---|---|---|---|
| | Unstructured | 133.64 | - | 132.12 | - |
| | SRigL | 135.12 | - (Baseline) | 131.42 | - (Baseline) |
| GPT-2 Small | + FixedRandPerm | 136.91 | **+1.32%** | 132.65 | **+0.94%** |
| | + PA-DST | 148.23 | **+9.70%** | 143.19 | **+8.96%** |

## C.3 EXPERIMENT DETAILS

All experiments are conducted on the NVIDIA Tesla A100 GPUs with the following configuration:

- Model: NVIDIA A100 40GB

Table 4: Memory overhead of permutation methods for ViT-B/16 with Diagonal sparsity on ImageNet-1K. The overhead percentage is calculated relative to the DynaDiag method.

| Model | Method | Memory (GB) @ 90% | % Overhead | Memory (GB) @ 95% | % Overhead |
|---|---|---|---|---|---|
| | Unstructured | 112.64 | - | 112.64 | - |
| | DynaDiag | 114.50 | - (Baseline) | 111.43 | - (Baseline) |
| ViT-B/16 (Diag) | + PA-DST | 138.38 | **+20.86%** | 135.01 | **+21.16%** |
| | + KaleidoscopePerm | 120.67 | **+5.39%** | 117.43 | **+5.38%** |

Table 5: Time and memory overhead of permutation methods for GPT-2 Medium with Diagonal sparsity on WikiText-103. The overhead percentage is calculated relative to the DynaDiag method.

| Model | Method | Sparsity @ 60% | | | | Sparsity @ 80% | | | |
|---|---|---|---|---|---|---|---|---|---|
| | | Time (h) | % Overhead | Memory (GB) | % Overhead | Time (h) | % Overhead | Memory (GB) | % Overhead |
| | Unstructured | 36.43 | - | 152.34 | - | 36.32 | - | 152.32 | - |
| | DynaDiag | 33.21 | - (Base) | 149.64 | - (Base) | 29.82 | - (Base) | 142.82 | - (Base) |
| GPT-2 Medium | + FixedRandPerm | 33.39 | +0.54% | 149.41 | -0.15% | 29.23 | -1.98% | 144.09 | +0.89% |
| | + PA-DST | 37.49 | **+12.88%** | 156.17 | +4.36% | 34.28 | **+14.95%** | 145.41 | +1.81% |
| | + KaleidoscopePerm | 41.98 | **+26.41%** | 155.27 | +3.76% | 37.98 | **+27.36%** | 137.43 | -3.77% |

- Memory: 40GB HBM2e

- Memory Bandwidth: $\sim$2.0 TB/s (higher than the 40GB version)

- TDP : 400W (PCIe: 300W)

- Peak FP32 Performance: $\sim$19.5 TFLOPS (same as 40GB)

- Peak FP16 Performance: $\sim$312 TFLOPS (same as 40GB)

### C.3.1 DATASETS

1. **ImageNet-1K** Deng et al. (2009) covers 1,000 object classes, with 1.28M training, 50,000 validation, and 100,000 test images. Images are typically resized and cropped to $224 \times 224$ for processing.

2. **WikiText-103** Merity et al. (2016) comprises over 100 million tokens extracted from verified Wikipedia articles. It is significantly larger than other language datasets, such as Penn Treebank (PTB) Marcus et al. (1993).

Table 6: Configuration of the CIFAR10 and CIFAR100 experiments with MLPMixer.

| Parameter | Value | Parameter | Value |
|---|---|---|---|
| Adam $\beta_1$ | 0.9 | Hidden | 128 |
| Adam $\beta_2$ | 0.99 | (Initial LR, Final LR) | $(10^{-3}, 10^{-6})$ |
| AutoAugment | True | Label Smoothing | 0.1 |
| Batch Size | 128 | Layers | 8 |
| CutMix Probability | 0.5 | LR Scheduler | Cosine |
| CutMix $\beta$ | 1.0 | Optimizer | Adam |
| Dropout | 0.0 | Random Seed | 3407 |
| Epochs | 300 | Weight Decay | $5 \times 10^{-5}$ |
| Hidden_C | 512 | Warmup | 5 epochs |
| Hidden_S | 64 | | |

Table 7: Configuration of the ImageNet experiments with ViT-Base and MLPMixer.Here X represents any of the sparse training methods that train a ViT-Base network.

| Model | Optimizer | Weight Decay | Learning Rate | Drop Path | Warmup/Epoch |
|---|---|---|---|---|---|
| ViT-Base | AdamW | 0.05 | 0.001 | 0.1 | 5/300 |
| X-ViT-Base | AdamW | 0.05 | 0.001 | 0 | 5/300 |
| Mixer-Small | AdamW | 0.1 | 0.001 | 0.1 | 5/300 |
| X-Mixer-Small | AdamW | 0.1 | 0.001 | 0 | 5/300 |

Table 8: Configuration of the ImageNet experiments with ViT-Large and Huge.

| Parameter | Value | Parameter | Value |
|---|---|---|---|
| Batch size | 256 | Horizontal flip | ✓ |
| Optimizer | AdamW | Random Resized Crop (RRC) | ✓ |
| Learning Rate (LR) | $3 \times 10^{-3}$ | Rand Augment | ✗ |
| LR decay | cosine | 3 Augment (ours) | ✓ |
| Weight decay | 0.02 | LayerScale | ✓ |
| Warmup epochs | 5 | Mixup $\alpha$ | 0.8 |
| Label smoothing $\varepsilon$ | 0.1 | Cutmix $\alpha$ | 1.0 |
| Dropout | ✗ | Erasing prob. | ✗ |
| Stochastic Depth | ✓ | ColorJitter | 0.3 |
| Repeated Aug | ✓ | Test crop ratio | 1.0 |
| Gradient Clipping | 1.0 | Loss | BCE |

Table 9: Configuration of the Wikitext-103 experiments GPT-2Small experiments.

| Model | Optimizer | Weight Decay | Learning Rate | Dropout | Warmup/Epoch |
|---|---|---|---|---|---|
| GPT-2-Small | AdamW | 0.1 | 0.0001 | 0.1 | 5/100 |
| DynaDiag | AdamW | 0.1 | 0.0001 | 0.1 | 5/100 |

## C.4 RAW VALUES

Table 10: Top-1 accuracy of structured sparse training methods at varying sparsities for a ViT-B/16 on ImageNet-1K. The results shown are from three runs for each data point. We see that there is no significant difference between the generalization performance of networks with learnt row and col permutation.

| Method | Perm. | 60% | 70% | 80% | 90% | 95% |
|--------|-------|-----|-----|-----|-----|-----|
| | | *dense accuracy = 78.5* | | | | |
| SRigL | Col | $78.04 \pm 0.011$ | $78.02 \pm 0.008$ | $77.83 \pm 0.009$ | $76.16 \pm 0.007$ | $69.24 \pm 0.008$ |
| PixelatedBFly | Col | $78.10 \pm 0.005$ | $78.04 \pm 0.007$ | $77.49 \pm 0.007$ | $74.09 \pm 0.008$ | $62.82 \pm 0.006$ |
| DSB | Col | $78.11 \pm 0.009$ | $77.95 \pm 0.008$ | $76.34 \pm 0.005$ | $73.09 \pm 0.004$ | $64.49 \pm 0.004$ |
| DynaDiag | Col | $78.53 \pm 0.007$ | $78.26 \pm 0.003$ | $77.85 \pm 0.005$ | $77.19 \pm 0.004$ | $70.12 \pm 0.007$ |
| SRigL | Row | $78.03 \pm 0.007$ | $78.02 \pm 0.001$ | $77.83 \pm 0.007$ | $76.16 \pm 0.005$ | $69.24 \pm 0.007$ |
| PixelatedBFly | Row | $78.12 \pm 0.004$ | $78.04 \pm 0.011$ | $77.49 \pm 0.003$ | $74.09 \pm 0.002$ | $62.82 \pm 0.009$ |
| DSB | Row | $78.09 \pm 0.004$ | $77.95 \pm 0.005$ | $76.34 \pm 0.003$ | $73.09 \pm 0.004$ | $64.49 \pm 0.009$ |
| DynaDiag | Row | $78.54 \pm 0.006$ | $78.26 \pm 0.008$ | $77.85 \pm 0.006$ | $77.19 \pm 0.003$ | $70.12 \pm 0.010$ |

## C.5 DETAILS OF LAYERS SPARSIFIED

In our ViT-B/16 experiments, we applied sparsity to the initial patch projection layer, the MLP layers, and the output projections of the multi-head attention (MHA) modules. For the GPT models, we sparsified all attention and MLP layers.

# D USE OF AI

We acknowledge the use of Google's Gemini and ChatGPT for assistance in editing the manuscript. The tools were used to refine sentence structure, correct grammatical errors, and improve the overall readability of the text. The intellectual contributions, including all methodologies, analyses, and conclusions, are solely those of the authors, who bear full responsibility for the final content of this work.

Table 11: Top-1 accuracy of sparse training methods at varying sparsities. We bold results that are not significantly different (based on paired asymptotic McNemar tests ($\alpha = 0.05$)) from the best-performing method (marked with a *) in each column. We see an increase in the generalization performance of all the structured sparse networks on the ImageNet-1K dataset.

| Model | Method | Perm. | Struc. | 60% | 70% | 80% | 90% | 95% |
|-------|--------|-------|--------|-----|-----|-----|-----|-----|
| | | | | *dense accuracy = 78.5* | | | | |
| | RigL | - | no | **79.75** | **79.28** | **78.71** | 77.24 | **71.50** |
| | SET | - | no | 78.15 | 78.01 | 77.78 | **77.01** | **71.48** |
| | CHT | - | no | **79.78** | **79.37** | 79.06* | 77.66* | 71.68* |
| | CHTs | - | no | 79.88* | 79.38* | **79.05** | **77.54** | **71.61** |
| | Mest | - | no | 78.04 | 77.76 | 77.39 | 76.45 | 69.67 |
| | SRigl | - | **yes** | 77.79 | 77.84 | 77.35 | 75.90 | 68.70 |
| | PixelatedBFly | - | **yes** | 78.04 | 77.90 | 77.31 | 73.89 | 62.52 |
| | DSB | - | **yes** | 77.98 | 77.85 | 76.26 | 72.89 | 64.17 |
| ViT-B/16 | DynaDiag | - | **yes** | 78.29 | 77.94 | 77.62 | **76.91** | 69.54 |
| | SRigL | Random | **yes** | 77.95 | 77.81 | 77.31 | 75.69 | 68.74 |
| | PixelatedBFly | Random | **yes** | 77.91 | 77.94 | 77.34 | 73.93 | 62.45 |
| | DSB | Random | **yes** | 78.06 | 77.84 | 76.27 | 72.84 | 64.23 |
| | DynaDiag | Random | **yes** | 78.21 | 77.92 | 77.67 | 76.93 | 69.54 |
| | SRigL | PA-DST | **yes** | 78.04 | 78.02 | 77.83 | 76.16 | 69.24 |
| | PixelatedBFly | PA-DST | **yes** | 78.10 | **78.04** | 77.49 | 74.09 | 62.82 |
| | DSB | PA-DST | **yes** | 78.11 | 77.95 | 76.34 | 73.09 | 64.49 |
| | DynaDiag | PA-DST | **yes** | **78.53** | **78.26** | 77.85 | **77.19** | 70.12 |
| | | | | *dense accuracy = 82.2* | | | | |
| | RigL | - | no | 81.85* | 81.57* | 81.70* | 78.26* | 72.11* |
| | SRigL | - | yes | 79.87 | 78.94 | 77.54 | 75.46 | 66.68 |
| | PixelatedBFly | - | yes | 79.13 | 79.06 | 79.33 | 75.12 | 66.59 |
| | DSB | - | yes | 79.44 | 77.46 | 75.34 | 73.55 | 66.77 |
| | DynaDiag | - | yes | 81.52 | 81.46 | 81.37 | 77.74 | 69.59 |
| | SRigL | Random | yes | 79.94 | 78.95 | 77.55 | 75.56 | 66.74 |
| ViT-L/16 | PixelatedBFly | Random | yes | 79.14 | 79.19 | 79.34 | 75.29 | 66.70 |
| | DSB | Random | yes | 79.42 | 77.51 | 75.39 | 73.71 | 66.74 |
| | DynaDiag | Random | yes | 80.21 | 80.16 | 79.52 | 75.31 | 69.56 |
| | SRigL | PA-DST | yes | 80.29 | 79.63 | 77.96 | 76.78 | 66.86 |
| | PixelatedBFly | PA-DST | yes | 79.33 | 79.23 | **80.33** | 75.67 | 66.79 |
| | DSB | PA-DST | yes | 79.59 | 77.61 | 75.51 | 73.79 | 66.91 |
| | DynaDiag | PA-DST | yes | **81.66** | **81.59** | **81.49** | 77.96 | 70.16 |
| | | | | *dense accuracy = 72.4* | | | | |
| | RigL | - | no | 73.21* | 73.23* | 73.47* | 73.01* | 70.41* |
| | SRigL | - | yes | 71.89 | 72.05 | 71.71 | 70.21 | 66.87 |
| | PixelatedBFly | - | yes | 71.95 | 71.91 | 71.45 | 69.17 | 67.85 |
| | DSB | - | yes | 69.94 | 70.21 | 68.90 | 65.16 | 60.88 |
| | DynaDiag | - | yes | **72.92** | 72.95 | **73.05** | **72.31** | **68.89** |
| | SRigL | Random | yes | **71.84** | 72.07 | 71.74 | 70.21 | 66.86 |
| Mixer-S/16 | PixelatedBFly | Random | yes | 71.91 | 71.94 | 71.49 | 69.17 | 67.91 |
| | DSB | Random | yes | 69.93 | 70.23 | 68.81 | 65.16 | 60.81 |
| | DynaDiag | Random | yes | **72.93** | 72.41 | 72.32 | 71.91 | 68.01 |
| | SRigL | PA-DST | yes | **72.56** | **72.91** | **72.79** | 71.24 | 67.16 |
| | PixelatedBFly | PA-DST | yes | 72.14 | 72.09 | 71.86 | 69.47 | **68.71** |
| | DSB | PA-DST | yes | 70.22 | 70.39 | 69.12 | 65.16 | 61.08 |
| | DynaDiag | PA-DST | yes | **73.09** | **73.11** | **73.21** | **72.51** | **69.19** |

Table 12: Perplexity of sparse training methods at varying levels of sparsity. We bold results that are not significantly different from the best-performing method (marked with a *) based on paired asymptotic McNemar tests ($\alpha = 0.05$). We see an improvement in the PPL (lower the better) score for all structured sparse training methods with permutaitons on the WikiText-103 dataset.

| Model | Method | Perm. | 40% | 50% | 60% | 80% | 90% |
|-------|--------|-------|-----|-----|-----|-----|-----|
| | | | *dense PPL = 22.21* | | | | |
| | RigL | - | 22.34* | **22.80** | 23.79* | 29.87* | 53.76* |
| | SRigL | - | 22.74 | 23.19 | 25.09 | 31.08 | 62.55 |
| | PixelatedBFly | - | **22.50** | 23.25 | 25.98 | 34.89 | 66.44 |
| | DynaDiag | - | 22.60 | **22.74** | **24.67** | **30.46** | 56.33 |
| GPT2-S | SRigL | Random | 22.70 | 23.19 | 25.21 | 31.11 | 62.42 |
| | PixelatedBFly | Random | 22.54 | 23.22 | 26.09 | 34.84 | 66.46 |
| | DynaDiag | Random | 22.61 | 23.19 | 25.12 | 31.69 | 57.61 |
| | SRigL | PA-DST | **22.41** | **22.94** | **24.59** | **30.40** | 60.96 |
| | PixelatedBFly | PA-DST | **22.41** | 23.01 | 25.69 | 34.71 | 66.06 |
| | DynaDiag | PA-DST | **22.44** | 22.69* | **24.51** | **30.26** | **55.49** |
| | | | *dense PPL = 20.18* | | | | |
| | RigL | - | 20.45* | 21.60* | 23.49* | 28.87* | 51.76* |
| | SRigL | - | 21.14 | 22.59 | 26.09 | 32.16 | 55.66 |
| | PixelatedBFly | - | **20.86** | 22.49 | 25.45 | 34.24 | 56.09 |
| | DynaDiag | - | **20.69** | **22.14** | **24.98** | **29.65** | 54.87 |
| GPT2-M | SRigL | Random | 21.19 | 22.55 | 26.01 | 32.19 | 55.69 |
| | PixelatedBFly | Random | 20.90 | 22.51 | 25.44 | 34.22 | 56.01 |
| | DynaDiag | Random | 21.65 | 22.67 | 25.17 | 30.39 | 54.81 |
| | SRigL | PA-DST | **20.57** | **22.20** | **25.04** | **29.89** | 55.13 |
| | PixelatedBFly | PA-DST | **20.69** | **22.23** | 25.31 | 33.19 | 55.71 |
| | DynaDiag | PA-DST | **20.55** | **21.91** | **24.71** | **29.21** | 54.26 |