# OpenReview forum: "Efficient Dynamic Structured Sparse Training with Learned Shuffles"
_ICLR.cc/2026/Conference — Submitted to ICLR 2026_

### Official Review · Reviewer_uGDU · 2025-10-30

**Soundness:** 3
**Presentation:** 3
**Contribution:** 2
**Rating:** 2
**Confidence:** 3

**Summary:**

This paper introduces Permutation-Augmented Dynamic Sparse Training (PA-DST), which augments structured dynamic sparse training with per-layer learned permutations to recover expressivity while preserving structured kernels at inference via index reordering. The authors provide extensive analysis and experiments, showing that the proposed PA-DST approaches unstructured DST accuracy at high sparsity (90–95%) while retaining the structured implementation’s performance.

**Strengths:**

1. The authors show that adding a permutation matrix to the structured sparse layers can greatly restore the loss of expressive power caused by sparsity while maintaining hardware friendliness.


2. The experiments span vision and large language models, multiple structured DST methods, and compare both accuracy and execution time overhead.


3. The paper is easy to follow. The introduction and motivation are clear and well-organized.

**Weaknesses:**

1. Lack of explanation of how the permutation matrix works in the backward phase and how the soft permutation is transformed to the permutation matrix.


2. The authors propose a method to improve model accuracy by making the sparse weights denser. However, in some cases, the values of these weights may differ significantly from the original dense weights. Should more rigorous and comprehensive experiments (larger datasets? Larger models? CNN?) be conducted to demonstrate the generalizability of the proposed method?


3. Figure 3 should be emphasized; it is hard to compare the performance improvement between the original method and the proposed PA-DST method.

**Questions:**

1. For the block DST, what does the permutation matrix look like? If a new DST sparse structure is proposed, what changes would be needed in PA-DST to adapt to the new structure?


2. The authors only discuss the ablation study of accuracy with row and column permutations in Table 10. However, accessing the weight matrix is typically done in either row-major or column-major. The impact of different access orders on computation time needs to be considered during implementation. It is hoped that the authors can provide an explanation for it.

---

> ### Author Response · Authors · 2025-12-03
> **Response to reviewer**
>
> We thank the reviewer for the feedback on our work. We address their concerns below.
>
> **W1:** The permutations in the backward pass are just used as another matrix and results in an additional matrix multiplication operation during training. However, after the initial phase, once we have settled on the permutation matrices, there is no additional operation in the backward pass. We apply the final permutation matrix to the weights during the forward pass and allow the training to proceed normally.
>
> **Q1:** Using our PA-DST approach on a new proposed method for sparse training just requires adding the loss corresponding to the permutation matrices to the existing task loss (as explained in Section 4).
>
> **Q2:**  This is a good point made by the reviewer. But during inference since the permutations are known you can arrange the data in the memory pre-emptively to get the best bandwidth.

---

### Official Review · Reviewer_BguH · 2025-10-30

**Soundness:** 3
**Presentation:** 3
**Contribution:** 2
**Rating:** 6
**Confidence:** 4

**Summary:**

This paper proposes PA-DST, a structured dynamic sparse training method by learning a permutation at each layer to restore lost expressivity. The approach maintains the efficiency of structured sparse kernels at inference by implementing permutations through lightweight index reordering. Experiments indicate that PA-DST can narrow the accuracy gap to unstructured DST at high sparsity levels while preserving the performance benefits of existing structured methods.

**Strengths:**

1. The authors present PA-DST, a general layer formulation that combines structured sparsity with a single learnt permutation matrix.

2. The authors prove tight combinatorial bounds showing that the added permutation recovers the expressivity lost by structured sparsity as compared to dense and unstructured sparse models.

3. Experiments demonstrate that with learnt permutations, structured sparse methods can achieve unstructured level accuracy across both vision and language tasks, while achieving significant speedup.

**Weaknesses:**

1. It isn't easy to compare the inference and training time of these structured sparse methods before and after using PA-DST.

2. Comparing the execution time on only one model (ViT-B/16) makes the experiment seem insufficient. Although Table 5 lists the execution time of GPT-2 Medium, it only shows the diagonal sparsity of 60% and 80%. What would it be like to compare inference and training time on larger models?

3. The author mentions in the paper that the proposed permutation-structured family can be accelerated in both forward and backward passes, but why is the difference in acceleration between inference and training so significant?

4. The subsection number should be corrected. e.g. 6.1.1 PPL vs Sparsity.

**Questions:**

1. In the ablation study, the authors show that there is no significant difference in accuracy between row and col permutations for ViT-B/16 model on ImageNet-1K. What are the results on other models and datasets? Are there differences in execution time for row and col permutations?

2. Section 6.3 shows the distance of permutations. What is the significance of doing this?

---

> ### Author Response · Authors · 2025-12-03
> **Response to reviewer**
>
> We thank the reviewer for the feedback on our work. We address their concerns below.
>
> **W1:** For clarity purposes we provide the training and inference times of the models in the table below:
>
> **Table: ViT-B/16 training time at different target sparsity levels.**
>
> | Method | 60% | 70% | 80% | 90% | 95% |
> | :--- | :---: | :---: | :---: | :---: | :---: |
> | Dense | 18.25 | 18.25 | 18.25 | 18.25 | 18.20 |
> | RigL | 18.11 | 18.10 | 18.08 | 18.11 | 16.73 |
> | SET | 18.13 | 18.13 | 18.07 | 18.09 | 16.83 |
> | CHT | 18.18 | 18.11 | 18.13 | 18.11 | 16.25 |
> | CHTs | 18.13 | 18.11 | 18.08 | 17.94 | 15.47 |
> | MEST | 18.14 | 18.11 | 18.21 | 18.15 | 16.97 |
> | SRigL | 18.12 | 17.77 | 15.69 | 14.85 | 12.33 |
> | PBFly | 17.978 | 16.88 | 15.24 | 13.68 | 11.98 |
> | DSB | 18.21 | 17.98 | 16.54 | 14.66 | 12.03 |
> | DynaDiag | 18.18 | 17.87 | 14.55 | 11.42 | 9.27 |
> | SRigL + PA-DST | 21.05 | 19.97 | 17.58 | 17.16 | 15.36 |
> | PBFly + PA-DST | 21.21 | 19.15 | 18.16 | 15.94 | 14.23 |
> | DSB + PA-DST | 21.45 | 19.61 | 18.45 | 16.11 | 14.31 |
> | DynaDiag + PA-DST | 21.05 | 19.88 | 17.41 | 14.51 | 12.51 |
>
> **Table: ViT-B/16 inference time at different target sparsity levels.**
>
> | Method | 60% | 70% | 80% | 90% | 95% |
> | :--- | :---: | :---: | :---: | :---: | :---: |
> | Dense | 151.20 | 151.20 | 151.20 | 151.20 | 151.20 |
> | RigL | 150.34 | 151.64 | 147.69 | 149.10 | 128.24 |
> | SET | 149.39 | 148.07 | 148.10 | 147.10 | 127.16 |
> | CHT | 147.21 | 146.59 | 145.27 | 143.20 | 121.13 |
> | CHTs | 146.07 | 145.13 | 144.31 | 141.10 | 115.29 |
> | MEST | 151.27 | 150.13 | 149.90 | 149.80 | 127.33 |
> | SRigL | 142.14 | 136.47 | 125.47 | 75.41 | 59.31 |
> | PBFly | 130.66 | 111.23 | 81.69 | 56.29 | 44.24 |
> | DSB | 132.46 | 113.45 | 87.98 | 53.22 | 42.11 |
> | DynaDiag | 109.87 | 87.87 | 69.47 | 48.14 | 32.31 |
> | SRigL + PA-DST | 146.22 | 141.19 | 131.21 | 78.91 | 64.72 |
> | PBFly + PA-DST | 133.24 | 115.15 | 83.94 | 60.12 | 47.03 |
> | DSB + PA-DST | 136.06 | 117.09 | 90.15 | 56.03 | 45.96 |
> | DynaDiag + PA-DST | 113.01 | 91.14 | 72.11 | 52.08 | 35.12 |
>
> **W3:** As we note in Section 4, the learnt permutations incur minimum overhead during inference as the permutations translate to a reindexing operation. However, during training, since we have soft permutation matrices for the initial phase of training, it results in additional matrix multiplication operations that are the main cause of the overhead and hence the difference.  We would also like to clarify that methods trained with PA-DST will have backward pass accelerated only if the original sparse structure supports backward acceleration. We elaborate on this in our response to reviewer aUjB (Please see W2 in Minor concerns).
>
> **Q1:** We do not observe any statistically significant difference in the inference times of models when using row or column permutations. However, we do find that for training purposes, it is better to use col permutations as the kernels can be fused which results in less slowdown as compared to the case of row permutation.
>
> **Q2:**  We wanted to look at the learnt permutations and one way of doing that is by measuring how far they are from say an identity matrix, which represents the case of no permutation. This means that if the distance is low, then using permutations really didn’t affect the specific layer, whereas if the distance is large then the permutations were useful for the corresponding layer.

---

### Official Review · Reviewer_aUjB · 2025-10-31

**Soundness:** 3
**Presentation:** 3
**Contribution:** 2
**Rating:** 6
**Confidence:** 5

**Summary:**

This paper proposes PA-DST, a method to augment dynamic sparse training by jointly learning permutations along with semi-structured sparse weight tensors. The input is multiplied by the permutations to enable semi-structured sparse topologies to be more expressive as measured by the number of linear regions in the functional output space of the sparse neural network (SNN). During training, a doubly-stochastic soft permutation matrix is used to learn the optimal per-tensor permutation. During inference, the soft permutation can be converted to a relatively efficient indexing operation applied to each layer's input. The empirical results suggest modestly improved model accuracy over existing semi-structured DST methods; however, the learned permutations do result in memory and latency overhead increases.

**Strengths:**

* The proposed method is novel and does not appear to have a direct corollary in the DST literature.
* The theoretical performance benefits of weight sparsity cannot be realized on commodity accelerators without imposing some form of structure on the sparse topology. As such, methods such as PA-DST which improve the expressibility of semi-structure sparse networks are crucial to creating practical SNNs for real-world tasks.
* The proposed method is well motivated through an analysis of the number of linear regions (NLR) present in dense, unstructured sparse, and semi-structured sparse neural networks.
* PA-DST universally improves the accuracy of existing semi-structured DST methods.
* Experimental results are conducted on both image classification and language modelling tasks.
* The paper includes all necessary hyperparameters required to reproduce the results.
* The proposed method is closed under transposition, enabling acceleration of backwards passes when the underlying structure allows it.

**Weaknesses:**

# Major concerns
The following represent key weaknesses that must be addressed to increase the rating:
* Potential for overfitting on WikiText: WikiText is a relatively small dataset. Based on Table 9, it appears that the GPT-2 models were trained for 100 epochs for a total of ~10B tokens. Generally, LLMs are pretrained on more diverse data for fewer (1-2 epochs, if any). Further, DST often achieves the highest accuracies when trained for more training steps than dense (i.e., RigL x5 steps ~= Dense x1). Pretraining on a larger dataset (eg., c4) for fewer epochs (0-1 ideally) would be more indicative of real-world pretraining runs and help establish whether PA-DST remains accurate when more diverse data is used.
* Memory overhead: At practical sparsities (50, 60%) that do not incur substantial PPL increases for GPT-2, the memory overhead of PA-DST is ~10-13%. Assuming the analysis was conducted with 16-bit weights and activations, the actual overhead in a deployment scenario may be double the value listed as W8A8 and even lower precision types become the norm for inference.
* Modest accuracy gains: While PA-DST provides significant gains in high-sparsity regimes (e.g., +1.32% for ViT-L/16 SRigL at 90%), for many other practical scenarios, the accuracy gains are more modest (<= 0.5%), which may not justify the additional memory and latency overhead.
* Additional training complexity: The learned permutations introduce a new hyperparameter, $\delta$ for the  permutation learning schedule soft/hard transition threshold. It is unclear if tuning this value may be required for other models and datasets.
* Lack of downstream task language evaluations: PPL has previously been shown to be a misleading metric when evaluating compressed LLMs.

# Minor concerns
The following are minor concerns, typos, etc. which would improve the work but do not affect the final rating:
* Scratch training vs. SFT: The proposed method is intended for training from scratch. Extending the experimental results to include supervised fine-tuning of pretrained models may improve the impact of the work.
* Clarify backwards pass acceleration: On L51, the authors note that the permutation-structure class is closed under transposition, thus “enabling sparse-to-sparse acceleration in both the forwards and backwards passes”. My understanding is that such acceleration would only be possible only if $S_l$ is in the set of sparse structures that offer exploitable structure for acceleration for both $S_l$ and $S_l^T$. I.e., SRigL’s fixed fan-in constraint allows for acceleration in the forwards, but not backwards pass, unless the constraint is also applied to the fan-out. As such, it may be more clear to state that the permutations do not prohibit acceleration of the backwards pass provided that the underlying sparse structure is itself amenable to transposition.

**Questions:**

* Do the benefits of PA-DST hold when pretraining is conducted on a more diverse dataset for less epochs?
* What dtypes are used for weights and activations for the training and inference benchmarks? How would lower bit-width types change the overhead analysis?
* Are the accuracy gains of PA-DST stable under multiple seeds? For instance, what is the standard deviation of select DST with and without PA-DST augmentation on ImageNet?
* How was the permutation learning schedule threshold tuned? How was the value of 0.22 selected? How sensitive is PA-DST to this value? Was 0.22 used for both the GPT-2 and image classification tasks? How many tokens does it take for the GPT-2 permutation to reach the penalty threshold?
*  When the sparse GPT-2 models are evaluated on a standard mutli-shot QA benchmark such as MMLU or OpenBookQA, are the benefits of learned permutations still apparent?
*  The authors note that they were unable to reproduce the SRigL inference benchmarks “due to lack of support in PyTorch”. SRigL’s code base includes CUDA kernels with pytorch bindings. Could the authors expand on the difficulties they encountered while trying to reproduce these results?

---

> ### Author Response · Authors · 2025-12-03
> **Response to reviewer**
>
> We thank the reviewer for their detailed feedback and appreciating our work. Please read our response below.
>
> **Major Concerns:**
>
> **W1:** We agree with the reviewer that the models we have trained have a potential of overfitting on Wikitext. Due to time and resource constraints, we were unable to run new experiments on GPT models for the rebuttal, but we plan to understand if that is the case in our revised version.
>
> **W2:** We acknowledge the overheads associated with the method. The overheads are primarily due to the complexity of learning the permutations. At the start, the permutation matrices are soft-permutation, so the complexity of the method is $O(n^2)$. We did try the Kaleidoscope [1] approach to learn permutation with just $O(nlogn)$ cost, but we found that the training was unstable specially for the GPT models.
>
> **W4:** We agree that the permutation learning schedule introduces an additional hyperparameter controlling the soft-to-hard transition. The hyperparameter $/delta$ does require tuning for the task at hand.
>
> **Minor Concerns:**
>
> **W1:** We agree with the reviewer and thank them for this suggestion. Due to the limited resource constraints, it will be valuable to show if a permutation based approach can be useful for finetuning large models instead of training them from scratch. We intend to do this for our revised version of the manuscript.
>
> **W2:** Yes. Acceleration of the backward pass is only possible if $S_l$ is in the set of the sparse structures that are transposable and hence can be accelerated in the backward pass. We will make the language clear in the revised manuscript.
>
> **Q2:** We use W16A16 dtypes for training and inference. Reducing the bit width of dtypes will reduce the overhead associated with the method. Since our focus was on training, we did not experiment with different dtypes.
>
> **Q3:** Yes. We found that the accuracy gains with PA-DST are stable under multiple seeds. For example, over three runs on ViT-B/16 at 90\% sparsity, the standard deviation was $\pm 0.007$ on ImageNet with PA-DST.
>
> **Q4:** We monitor the permutation matrices loss during training to decide the threshold. We find that after a point, the loss stops decreasing and we use that point to set the threshold We used 0.22 for the vision tasks and 3.21 for GPT-2 Medium on Wikitext. We find that for a GPT-2 model, we needed training for upto 6B tokens to reach the threshold.
>
> **Q5:** We have not evaluated the models on benchmarks such as MMLU as we believe the size of the models is too small to get a meaningful baseline to compare the PA-DST against. However, we plan to carry out finetuning of large models using permutations and run those models on the downstream tasks in our revised version.
>
> **Q6:** We have elaborated on this in our response to reviewer KXyo.
>
>
> [1] Dao, Tri, et al. "Kaleidoscope: An efficient, learnable representation for all structured linear maps." arXiv preprint arXiv:2012.14966 (2020).

---

### Official Review · Reviewer_KXyo · 2025-11-01

**Soundness:** 1
**Presentation:** 2
**Contribution:** 4
**Rating:** 2
**Confidence:** 4

**Summary:**

The authors propose a novel structured dynamics sparse training (DST) method that uses both a permutation matrix and structured weight matrix jointly to learn an expressive representation that is also amenable to acceleration in both the backwards and forwards pass on real-world hardware, and can exploit a variety of structured formats, e.g. N:M, block and diagonal. The authors claim their method improves generalization over existing structured DST baselines significantly at 90-95% sparsity. The authors evaluate their method on vision and language models, includes ImageNet-1K and WikiText-103 with models such as ViT-B/16 and GPT-2 respectively.

**Strengths:**

* The authors claim to accelerate both the backwards and forward pass, unlike all existing structured DST methods I'm aware of, which only accelerate the forward pass.
* The authors evaluate their method on a wide array of structured sparsity patterns, as opposed to only a fixed and specialized N:M sparsity pattern as in SRigL
* The authors evaluate on strong models/dataset baselines, comparable to what has been used in prior work.
* The proposed methodology is interesting and novel, in particular the permutation shuffling.
* The authors address concerns over the efficiency of implementation of their proposed methodology.

The authors also claim to have addressed two of the most significant gaps in the literature in my opinion: accelerating the backwards pass/training and accelerating DST with arbitrary N:M sparsity patterns. These are in my mind the strength of this paper and really piqued my interest in their work.

**Weaknesses:**

* The motivation the authors have chosen to write their story around, and indeed background/abstract is misleading: structured dynamic sparse training (DST) methods such as SRigL demonstrate that they can match the accuracy of unstructured DST methods except perhaps at extremely high sparsity (higher than the authors demonstrate here), and I believe most researchers in the field would not claim this is a significant gap.
* The results presented appear to be very different than those in the SRigL baseline paper (which is confusing as they later claim they can't run SRigL inference), and without any explanation as to why. In fact they appear to match the SRigL *without ablation* results instead (i.e. the results used to demonstrate why neuron ablation is required), and since the authors don't mention how they have implemented SRigL or found these numbers, I'm concerned this is where the claim of a large generalization gap is coming from, and perhaps why the introduction reads as so misleading if one is aware of the SRigL results.
* As has been established by much of the literature over the past 1-2 years, perplexity is a very poor evaluation alone of the relative performance of a language model performance post-pruning or compression, or indeed in any context. Unfortunately it is not sufficient to look at perplexity at this point, instead there do have to be some task evaluation benchmarks in the results. See for example "Compressing LLMs: The Truth is Rarely Pure and Never Simple, Jaiswal et al., 2024"
* The theoretical justification was hard to follow and not very convincing to me. More importantly it comes off to the reader as disconnected from the story which frankly is a very empirical claim (improved generalization) or engineering claim (accelerating backwards pass/arbitrary n:m sparsity). I don't understand why so much of the main paper was devoted to this, I would much rather have seen the more important details of the proposed methodology that are in the appendix - a section frankly most readers of your work are not required to read.
* As someone who is somewhat familiar with the SRigL codebase, I know it's written in *PyTorch*, so this statement is extremely worrying and confuses me as to how the authors are evaluating SRigL, which is the strongest baseline method, and main result in their improved generalization claim: "We estimate the execution times for SRigL as per their methodology and are unable to run end-to-end training and inference due to lack of support in PyTorch". It is generally expected that you are either replicating the results of the baseline or explicitly using comparable results from the baseline paper, I don't see how you can estimate real-world execution times, or give generalization results without being able to do inference.
* The method proposed suffers a significant memory and compute overhead unlike existing structured DST methods
* It is unclear what hardware (even if CPU or GPU or other) the timings in figure 3 are run on, or if these are indeed real-world timings at all given the author's comments in the paper, and if they include the compute overheads discussed for the proposed method. Given this I can't evaluate the claims on inference/training times at present.

Given the story of the paper is increased generalization, I have to evaluate the paper based on the claims around increased generalization rather than some of the more impressive aspects of the proposed algorithm, and the generalization results are unfortunately not well supported by the evidence, in particular the main (SRigL) baseline and confusion as to where those results come from.

Additional Feedback on Improving Paper:
* The authors also claim to have addressed two of the most significant gaps in the literature in my opinion: accelerating the backwards pass/training and accelerating DST with arbitrary N:M sparsity patterns. I can't understand why these aren't the main motivation and story of the paper, these are much more interesting than claiming a minor increase in generalization. Also the evidence to support these claims would be clear to evaluate.
* Figure 2 is extremely hard to read due to the very few colours and symbol overlap, especially on a printed version, but even on screen. Being such an important figure for the claims being made this is a big problem.

**Questions:**

* Are you using SRigL *with ablation* in your experiments? Is this the original codebase or your own code replicating the results? Where do the SRigL results in Figure 1 come from if you are unable to run training or inference for SRigL?
* Does your algorithm require neuron ablation as used in SRigL to match generalization at high sparsity?
* "We estimate the execution times for SRigL as per their methodology and are unable to run end-to-end training and inference due to lack of support in PyTorch". Explain this statement, why were you unable to use or replicate the results of SRigL (especially considering it is written in PyTorch)?
* Exactly where do the timings in Figure 3 come from? What hardware were they run on? GPU, CPU? Are they all timed the same way? Do the PA-DST timings include the compute overhead?
* Do you have any further evaluation that perplexity for the language results, e.g. on task benchmarks?

---

> ### Author Response · Authors · 2025-12-03
> **Response to reviewer**
>
> We thank the reviewer for their detailed feedback. It has helped us to do a fair comparison of methods. We address the concerns below.
>
> **W1:**  We agree that recent work such as SRigL has shown that structured DST can match the accuracy of unstructured DST method on vision tasks at high sparsities. Our motivation is more specific: we focus on large Transformer/LLM-scale models. In this domain, if our understanding is correct, recent works have shown that hardware friendly patterns still tend to incur accuracy drops as compared to unstructured sparsity [1]. However, we do acknowledge that the gap has been closing between structured and unstructured sparsity for larger models and our work is an attempt towards the same.
>
> **W2 & W4** We understand the confusion regarding our SRigL numbers. Let us clarify that in detail:
> We use the official repository to obtain the results on all the models. The official repository also contains the cuda kernels used to accelerate the SRigL method. However, at the time of paper submission we were unable to run the code on our A100 HBM2e and H100 GPUs. Therefore, don’t use the cuda kernels and for inference and training, we treat the model as dense with zeros. This ensures that for training and inference purposes the generalization results are obtained based on the official implementation.
>
> Our training and inference time numbers are estimates based on the timing results in the SRigL paper and adjusted for A100 GPUs flops.
>
> After some effort, we were able to get the cuda kernels to work on our available GPUs and we report below the generalization numbers with and without acceleration:
>
> **Table: ViT-B/16 Top-1 accuracy comparison (SRigL vs. SRigL with acceleration).**
>
> | Method | 60% | 70% | 80% | 90% | 95% |
> | :--- | :---: | :---: | :---: | :---: | :---: |
> | SRigL | 77.79 | 77.84 | 77.35 | 75.90 | 68.70 |
> | SRigL (with acceleration) | 77.81 | 77.82 | 77.39 | 75.94 | 68.73 |
>
> This shows that the generalization performance SRigL trained models doesn’t different drastically.
>
> We also report inference time number with accelerated SRigL implementation. We find that the estimates do differ from the actual end to end numbers and we aim to do a fair comparison in our revised manuscript:
>
> **Table: ViT-B/16 inferencetime comparison (SRigL vs. SRigL accelerated).**
>
> | Method | 60% | 70% | 80% | 90% | 95% |
> | :--- | :---: | :---: | :---: | :---: | :---: |
> | SRigL | 142.14 | 136.47 | 125.47 | 75.41 | 59.31 |
> | SRigL (accelerated) | 129.04 | 111.27 | 101.81 | 69.21 | 51.29 |
>
> **W3 & Q5:** We agree with the reviewer that just looking at PPL is probably not a good enough metric to understand the generalization performance of LLMs. However, for the models we have trained with limited resources, the baseline performance on the downstream tasks is poor. The largest model we train (GPT-2M) has just ~ 345M  parameters and on benchmarks such as MMLU, it doesn’t perform well. Therefore, we aim to extend our technique to larger models where instead of pre-training, we would like to finetune the models (as suggested by reviewer aUjB) and compare the performance on downstream tasks.
>
> **W6:** We acknowledge the overheads associated with our proposed method. Our aim with the work was to show the possible advantages of learning permutations during training. However, in our revised version of the work, we propose to try cheaper methods, such as the one proposed by  Droge et al. [2]  to learn permutations and possible reduce the overhead.
>
> **W7:** All our training and inference runs are on GPUs. We mention how we run the baselines in our experimental setup in Section 5. Our PA-DST results for training and inference include the overhead associated with the method.
>
> **Response to feedback:**
>
> We appreciate reviewers feedback on improving the flow of the paper. However, we would like to clarify that we do not claim to accelerate the backward pass for any sparsity pattern. Our method is plug and play when it comes to extending any current training method. So, if a sparsity pattern can be accelerated in the backward pass, PA-DST will not hinder with that. We would also like to highlight that there have been previous works that have shown how backward pass can be accelerated for the case of N:M sparsity [3][4].
>
>
> [1] Fang, Gongfan, et al. "Maskllm: Learnable semi-structured sparsity for large language models." Advances in Neural Information Processing Systems 37 (2024): 7736-7758.
>
> [2] Dröge, Hannah, et al. "Kissing to find a match: Efficient low-rank permutation representation." Advances in Neural Information Processing Systems 36 (2023): 48459-48471.
>
> [3] Hubara, Itay, et al. "Accelerated sparse neural training: A provable and efficient method to find n: m transposable masks." Advances in neural information processing systems 34 (2021): 21099-21111.
>
> [4] Zhang, Yuxin, et al. "Bi-directional masks for efficient n: M sparse training." International conference on machine learning. PMLR, 2023.

---

### Meta-Review · Area_Chair_AL5a · 2026-01-07

**Summary:**

This paper proposes a mechanism for _structured_ sparsity in a dynamic way, aiming to bridge the gap between DST (good performance) with structured sparsity (which is amenable to hardware acceleration).

While quite interesting and promising, there were a number of valid concerns raised:
1. Evaluation
  - There are concerns of possibly overfitting, which the authors acknowledged
  - The use of perplexity is not a good metric, which the authors acknowledged
  - There were valid concerns regarding the claimed "unusability" of the SRigL code. The authors claimed it did not run on their hardware, and it seems they reported a mixture of their own results along with data taken from the original paper (such as timing), which is not great.
2. Ambiguous wording
  - some reviewers were led to believe that backwards passes were accelerated, which turns out to not be the case
3. Clarity of presentation
  - figures 2 & 3, specifically

As such, I do not believe the paper is ready for publication. There were also some

**Reviewer Concerns:**

I highlight the main reviewer concerns below.

## KXyo
- W1 (gap between structured and DST is not as wide as claimed). The authors acknowledge this is the case: "we do acknowledge that the gap has been closing between structured and unstructured sparsity for larger models and our work is an attempt towards the same."
- W2 & W5 (discrepancy with SRigL code and the claim that the authors were "unable to run end-to-end training and inference due to lack of support in PyTorch", which is surprising given that SRigL code _is_ in PyTorch). The authors respond saying they did use the official repository but at the time it did not run on their hardware. Since then, they were able to run and provide some metrics, although it calls into question the validity of the main paper's results and its comparison to SRigL.
- W3 (perplexity is a very poor evaluation alone of the relative performance of a language model performance post-pruning or compression, or indeed in any context). The authors justify this choice with "for the models we have trained with limited resources, the baseline performance on the downstream tasks is poor", stating that in the future they'd like to evaluate with post-training. This is a good idea, but it suggests the paper as-is is not ready for publication yet.
- W4 (theoretical justification was hard to follow and not very convincing to me). Not really addressed by authors.
- W6 (significant memory and compute overhead unlike existing structured DST methods). The authors respond with "We acknowledge the overheads associated with our proposed method. Our aim with the work was to show the possible advantages of learning permutations during training. However, in our revised version of the work, we propose to try cheaper methods", which again suggests that in its current form, the paper is not ready for publication.

## aUjB
- W1 (overfitting on WikiText). The authors agree: "We agree with the reviewer that the models we have trained have a potential of overfitting on Wikitext. Due to time and resource constraints, we were unable to run new experiments on GPT models for the rebuttal, but we plan to understand if that is the case in our revised version.", which suggests the paper is not ready for publication in its current form.
- W2 (Memory overhead). "We acknowledge the overheads associated with the method". Related to discussion of W5 from reviewer KXyo.
- W3 (Modest accuracy gains, which may not justify the additional memory and latency overhead). Not addressed by authors.
- W4 (Additional training complexity). The authors agree "We agree that the permutation learning schedule introduces an additional hyperparameter controlling".
- W5 (PPL has previously been shown to be a misleading metric when evaluating compressed LLMs). Not addressed in the response to this reviewer, but they agreed with reviewer KXyo on this point, suggesting paper is not ready yet.
- The questions raised by this reviewer were mostly addressed by authors, except for the point about the PyTorch implementation of SRigL (see discussion on concerns from KXyo).

## BguH
- W1 (It isn't easy to compare the inference and training time of these structured sparse methods before and after using PA-DST). The authors respond to this adequately.
- W2 (What would it be like to compare inference and training time on larger models?). Not addressed
- W3 (why is the difference in acceleration between inference and training so significant?). This is related to the clarity of presentation, where most reviewers were led to believe that the authors were able to accelerate backward passes. "We would also like to clarify that methods trained with PA-DST will have backward pass accelerated only if the original sparse structure supports backward acceleration".

## uGDU
- W2 (Should more rigorous and comprehensive experiments (larger datasets? Larger models? CNN?) be conducted to demonstrate the generalizability of the proposed method?). Not addressed
- W3 (Figure 3 should be emphasized). The figure is very hard to read, as well as Figure 2. Not addressed.

**Reviewer Scores:**

- **KXyo:** Currently at 2, seems unlikely they would increase; if they did, they would likely not go above a 4.
- **aUjB:** Currently at a 6, highly unlikeky to increase and actually quite likely to _decrease_ their score, given the discussion with other reviewers.
- **BguH:** Currently at a 6, highly unlikeky to increase and actually quite likely to _decrease_ their score, given the discussion with other reviewers.
- **uGDU:** Currently at 2, unlikely to increase, or at most to a 4.

---

### Decision · Program_Chairs · 2026-01-26

Reject